# Is a Pleasant Policy Environment Conducive to Green Growth of Central China?

**DOI:** 10.3390/ijerph19137647

**Published:** 2022-06-22

**Authors:** Jianqing Zhang, Enze Gong, Fangheng Tong, Shuting Li

**Affiliations:** 1Institute for the Development of Central China, Wuhan University, Wuhan 430072, China; jqzhang@whu.edu.cn; 2School of Economics and Management, Wuhan University, Wuhan 430072, China; 2020201050129@whu.edu.cn (F.T.); 2019201050037@whu.edu.cn (S.L.)

**Keywords:** policy environment, green growth, Central China, empirical research, regulation capture

## Abstract

Improving the pleasantness of the policy environment in Central China stimulates economic growth, but it also contains higher risks of pollution. Based on the data of 80 cities in Central China from 2006 to 2018, the entropy method was used to estimate the pleasantness of policy environment in the region. How the policy environment has changed and whether the pleasant policy environment in the region is conducive to green growth were empirically studied. The results show the following: (1) The current attempts to improve the pleasantness of the policy environment in Central China is not conducive to green growth of the region. (2) Improving the pleasantness of the policy environment has indirect negative impacts on green growth through widening the development gap between prefecture-level cities and provincial capitals, as well as encouraging foreign trade; meanwhile, it also has an indirect positive impact through stimulating industrial diversification. The policy environment does not indirectly affect green growth through affecting technological innovation. (3) The policy environment in Central China will have a heterogeneous effect on the green growth of cities from the perspective of spatial heterogeneity and heterogeneity of city characteristics. In this paper, policy implications are proposed.

## 1. Introduction

Policy environment, which refers to whether the government’s intervention in a region is friendly to the operation of enterprises, is an important factor affecting industrial development in Central China. After benefiting from “The Third-Line Construction” movement in the 1960s and 1970s, Central China has built a relatively complete industrial system (Central China has established automobile manufacturing, metallurgy, chemical industry, food-processing industry, etc.). However, after 1978, the Chinese government’s economic focus has been shifted to the eastern coastal areas and western ethnic minority areas. The lack of industrial stimulating policies has made Central China a typical sub-developed region. From the perspective of industrial structure, there are relatively fewer types of industries in Central China; there are mainly a raw material industry, fuel power industry, and agricultural processing industry [1]. Relatively higher energy consumption and pollution often result in lowering the operating profits of these industries, meaning that there is a lower potential of long-term growth in this region. From the perspective of regional competition, both external competition and internal competition faced by Central China are becoming increasingly fierce. On the one hand, with the formation of “three bases and one hub”, the industrial competitiveness of Central China has risen sharply, and this will inevitably attract competition from developed eastern regions; on the other hand, a higher level of industrial isomorphism will induce fierce competition among cities within Central China. Meanwhile, participating in international trade by increasing pilot free-trade zones and expanding the influence radius of China Railway Express reduce the intensity of domestic competition for products, increase the tendency of industries to agglomerate to big cities, and further intensify competition among cities [2]. In order to cope with fierce competition and stimulate sustainable economic growth (Sustainable growth means maintaining the growth rate in future), governments in Central China began to spontaneously introduce industrial stimulating policies and continuously improve the level of public services. At the same time, the central government proposed the “Rise of Central China” strategy in 2006 to balance regional development, thus overcoming the policy shortcomings of industrial development in Central China from the national level (Data from the official website of the Ministry of Finance of China). With the return of the governments’ support, the economic growth of Central China has been effectively promoted by encouraging industrial policies represented by the “Two Comparative Implementation Policies” [3]. In 2018, the real GDP of Central China increased by about 4.53 times compared with 2006.

The pleasant policy environment fostered by these interventions may contain high risks of damaging the ecological environment in Central China, making it ineffective to promote regional green growth. The following can be illustrated from the composition of government’s intervention in Central China: Both encouraging policy and restrictive policy are frequently used by governments in emerging markets for their efficiency of creating advantages for industrial development [4]. From the perspective of encouraging policy, high-quality public service, subsidies, tax relief, or even land-transfer preference might be delivered to attracting polluting industries to achieving short-term growth. As mentioned above, the development of chemical, heavy, and resource industries in Central China has comparative advantages (Industries of this kind are expensive to close or transform for their huge fixed-asset investments and employment scale, so stimulating the development of these industries is the fastest way to achieve growth with the lowest costs). Encouraging the development of such industries can quickly lead to economic growth, but at the cost of increasing energy consumption and pollution emissions. From the perspective of restrictive policy, there is a high possibility for governments in Central China to relax environmental regulation to keep cities attractive to industries. Cities in Central China can hardly match eastern cities in terms of public services, technological innovation, and capital stock, so they can only increase their advantages by relaxing environmental regulation. That means that governments in sub-developed regions such as Central China may actively or passively “collaborate” with polluting industries due to fierce regional competition and huge pressure of growth, causing the problem of “environmental regulation capture”. On the other hand, slow reform of the administrative system in Central China results in defects of supervision over the bureaucratic system, making it easier to induce environmental corruptions. Kim and Wilson (1997) have confirmed that sub-developed regions face competition from both developed regions and other sub-developed regions, and this may lead to the phenomenon of environmental regulation “race to the bottom line” [5]. Liu and Li (2021) point out that some polluting enterprises in developed regions will move to sub-developed regions by disguising themselves as environmental-protection enterprises in the process of industrial transformation [6]. Therefore, improving the pleasantness of policy environment could make it more difficult to reconcile the contradiction between environmental protection and economic growth in Central China.

At present, ecological problems in Central China are prominent: Industrial waste water emission and industrial soot emission in Central China reached 2317.28 and 10,813.89 million tons, respectively, in 2018. A large number of pollution industries in Eastern China have been moving into Central China; the proportion of pollution-intensive industries in Central China shows an obvious upward trend and reached 30% in 2015. Mineral resources are drying up in Central China; even two regions of Shanxi Province have been selected as resource exhausted cities (Data can be found on the official websites of the National Bureau of Statistics and the Ministry of Natural Resources of China). The 14th “Five-Year Plan for the National Economic and Social Development of the People’s Republic of China and the Outline of Long-Term Goals for 2035” states that “governments should resolutely curb the blind development of high-energy-consuming and high-emission projects, and promote green transformation to achieve positive development” (Total contents: http://www.gov.cn/xinwen/2021-03/13/content_5592681.htm, accessed on 13 March 2021). It is urgent for Central China to improve its green economic efficiency. Since governments’ intervention is an un-substitutable driving force for the growth of Central China in the short term (Industries in China tend to agglomerate to economically developed regions; if there is no intervention from governments in Central China, the development gap will become wider between Central China and Eastern China. Industries in Central China are those with heavy investments and a long return cycle, which is less competitive compared to IT or financial industries mostly located in eastern large cities, making them more reliant upon policy support [1]), it is necessary to figure out whether the change of policy environment in Central China is conducive to green growth and in which way we can reduce its adverse impact on environmental protection. However, few studies have accurately estimated the pleasantness of policy environment in Central China, nor have they clarified its relationship with green growth.

Accordingly, the objective of this study was to empirically clarify whether and how improving the pleasantness of the policy environment in Central China affects regional green growth and then provide useful policy implications for sub-developed regions to reduce adverse effects of administration on environmental protection and explore a green growth path. To achieve this, firstly, the entropy method was adopted to quantitatively estimate the pleasantness of policy environment for industrial development based on the data from prefecture-level cities in Central China. Secondly, the spatiotemporal variation of policy environment in Central China and its possible reasons were analyzed. Thirdly, econometric models were constructed to test whether the pleasant policy environment in Central China is conducive to green growth and its possible impact mechanism. Lastly, a heterogeneity analysis was conducted to further explore whether the impact of policy environment on green growth is heterogeneous.

## 2. Literature Review

### 2.1. Research on Policy Environment for Industrial Development

It has come to a consensus that industrial policy can significantly affect economic growth and social development. The main debate at present on the research of policy environment lies in how to measure policy environment and whether industrial policy is efficient. Since a government’s expenditure can reflect a government’s investment in public services and infrastructure construction, some studies only use a single government’s expenditure to measure whether policy environment for industrial development is pleasant [7]. In these studies, the policy environment in developed regions is much more pleasant than that in developing regions. Compared with a single government’s expenditure indicator, a more common operation is to use multidimensional indicators such as a government’s responsibility, public services, and taxation burden to measure policy environment [8,9]. This kind of research takes encouraging policies as the measurement index of policy environment. To handle the pressure of environmental protection and industrial transformation, governments in China have intensively issued a series of restrictive policies to strengthen environmental regulation, such as chemical enterprises within 5 km along the Yangtze River are forced to “move, transfer or close”. Therefore, some studies use restrictive industrial policies, especially environmental regulation policies, to measure policy environment [10,11]. As for the effect of industrial policies, some scholars believe that a positive effect of industrial policy is significant on economic growth, so the government should actively participate in economic activities to create a pleasant policy environment for industrial development [12,13]. However, scholars such as Masu (1992) and Bernstein (1995) pointed out that the government could collude with the supervised entities for some purpose to invade public interests, resulting in the problem of “regulation capture” [14,15]. Li et al. (2021) have confirmed that there exists a problem of “regulation capture” in China’s environmental protection policies [9].

### 2.2. Research on Green Growth of China

Green growth is a hot topic in current research. Many scholars have evaluated the status of China’s green growth and deeply discussed possible influencing factors. The World Bank (2012) and OECD (2010) have defined the concept of green growth, stating that green growth is a sustainable economy, which essentially seeks the relative or absolute decoupling between economic growth and environmental degradation [16,17]. In research, the green growth index is often measured by green total factor productivity [18]. Kuang and Peng (2012) studied the change of China’s environmental production efficiency and total factor productivity from 1995 to 2009 and proposed that energy conservation and emission reduction led to the growth speed of environmental total factor productivity faster than that of traditional total factor productivity in normal years [19]. Yu and Wei (2021) estimated the green economic efficiency of prefecture level cities in China from 2004 to 2016 and also concluded that the green economic efficiency of eastern cities is higher than central and western cities [20].

Besides of measuring and commenting on the status of China’s green growth, more studies have explored its possible influencing factors. From the perspective of industrial development, on the one hand, domestic industries, including capital and talents, tend to agglomerate to big cities, and thus will have a direct impact on China’s green total factor productivity [21]; on the other hand, continuous industrial transformation and upgrading in China implies that the industrial structure is unstable, and this will have inter-provincially heterogeneous positive impacts on green total factor productivity [22,23]. From the perspective of opening to the world, participating in foreign trade, expanding the scale of FDI and OFDI, and increasing pilot free-trade zones can directly affect China’s green total factor productivity, under certain circumstances, the effect will be significantly positive [24,25,26]. From the perspective of technological innovation, Xu, Wang, and Chen (2020) have confirmed that different forms of technological innovation can significantly affect industrial green growth [27]. Jiang et al. (2021) pointed out that the construction of smart city had effectively improved China’s green total factor productivity. Compared with market factors affecting green total factor productivity, administrative factors, especially environmental regulation, are equally important [28]. Li and Bai (2020) state that the coordination of restrictive industrial policies and encouraging industrial policies improves green total factor productivity of manufacturing [29]. Based on the introduction of carbon emission trading policy, Zhang and Qiao (2021) confirmed that the intensify of environmental regulation can improve green total factor productivity [30].

### 2.3. Research on the Relationship between Policy Environment and Green Growth

There is no consensus on whether a pleasant policy environment can promote green growth. “Promotion theory” holds that government’s behavior can improve green growth from two aspects: On the one hand, the government can improve cities’ attraction to talents and high-quality enterprises through reducing the taxation burden, improving the quality of public services and infrastructure construction, and then promoting industrial upgrading [4,31]. On the other hand, restrictive policies such as environmental regulation can effectively supervise behavior of enterprises, reducing pollution emissions by stimulating the improvement of production technology and equipment of enterprises, and then have a positive direct impact on green total factor productivity [32]. “Inhibition theory” mainly explains the negative impact of government’s behavior on green growth from the perspective of restrictive policies. Because green growth contains connotations of “green” and “growth”, there are also two reasons for the negative impact: (1) Strengthening of environmental regulation has increased the cost of pollution control for enterprises and reduced the attraction of cities to industries, resulting in a direct adverse impact on economic growth and productivity improvement [33]. (2) Local governments, especially in developing areas, are often captured due to their disadvantage position and bureaucratic style, inducing the problem of “regulation capture” in environmental regulation which can cause obvious damage to the ecological environment [34,35]. “Threshold theory” points out that improving the pleasantness of the policy environment will have different effects on green growth under different conditions. For example, the effects could be inverted “N” type or “U” type [36,37].

### 2.4. Comments on Existing Research Studies

Recognizing the importance of policy environment in economic activities, existing research studies have conducted in-depth explorations of how to measure the policy environment and whether industrial policies will be conducive to green growth. However, these research studies have failed to clarify the changing trend of policy environment in Central China and also are unable to answer whether there is serious “regulation capture” behind the improvement of policy environment in Central China. From the perspective of policy environment measurement, we believe both restrictive and encouraging policies can affect the pleasantness of industrial development, whereas most studies fail to integrate these two kinds of policies into one policy environment index, resulting in the inaccurate evaluation of the pleasantness of policy environment in the existing literature. From the perspective of data type, most studies use national-level data, whereas governments in Central China are more easily “captured” since Central China is a sub-developed region with inflexible industrial structure and faces greater competitive pressure in the process of development, so conclusions drawn from researches at the national level are not necessarily applicable to Central China. Based on the existing research, this paper has made the following marginal innovations: Firstly, we used the entropy method to accurately estimation of policy environment pleasantness of central cities by considering both influence of restrictive and encouraging policies. Secondly, the relationship between the pleasant policy environment and green growth in Central China was theoretically and empirically analyzed. Lastly, implications to reduce the adverse impact of administration on green growth of the region were proposed.

## 3. Data Sources

This paper uses the data of 80 prefecture-level cities in Central China. Governments’ expenditure, fixed assets, public service data, pollution emission data, foreign investments, GDP of the city, employment, population, and financial data can be found in China City Statistical Yearbooks from 2007 to 2019. Energy consumption and capital stock can be found in China Statistical Yearbooks, Statistical Yearbooks of 6 provinces in Central China from 2007 to 2019. Foreign trade scale can be found in Statistical Bulletins of each city from 2006 to 2018. Data of patents are from CNRDS database and the official website of National Bureau of Statistics. Data of some cities are excluded: data of Chaohu, which has been incorporated into Hefei, are excluded; data of Tianmen, Xiantao, Qianjiang, and Jiyuan are excluded, for these cities are county-level cities governed by provincial governments; and data of Enshi and Xiangxi are excluded. Indicators such as income, expenditure, and cost are all deflated by the price index of corresponding year, removing the impact of inflation.

## 4. Spatiotemporal Variation of Pleasantness of Policy Environment in Central China

### 4.1. Important Events Affecting Policy Environment in Central China

From 2006 to 2018, there were three landmark events that changed the policy environment for industrial development in Central China. We list these three important events on the timeline of Figure 1. On 15 April 2006, the central government issued “Several Opinions of The CPC Central Committee and The State Council on Promoting the Rise of Central China”, marking how the “Rise of Central China” has officially become a national strategy. Before this opinion was issued, the central government’s economic focus was on the eastern coastal region and the western inland region, leading to the lack of policy support for industrial development in Central China and further causing policy environment for industrial development in Central China to be relatively unpleasant. In order to stimulate economic growth of Central China and balance regional development, “encouraging policies” have gradually increased since 2006 in this region. The central government has allocated special funds every year to implement the strategy of the “Rise of Central China” (After “Tax Sharing Reform”, only the central government in China has the authority to grant tax concessions. Specific preferential terms can be found in the “Two Comparative Implementation Policies” issued by the General Office of the State Council on 1 January 2007), given preferential tax and financial support to enterprises in Central China (Include the establishment of high-tech zones, bonded zones, free-trade zones, and other special zones in Central China for local governments to experiment economic stimulus policies), and given more autonomy to local governments to improve the policy environment for industrial development (According to the “13th Five Year Plan for Promoting the Rise of Central China” and annual reports of provincial and municipal governments in Central China). Local governments have introduced financial subsidies, preferential land-transfer prices, and better infrastructure to attract and cultivate industries. The contradiction between economic growth and environmental protection has become increasingly prominent, so the report of the 18th National Congress of the CPC in 2012 pointed out that we should “put the construction of ecological civilization in a prominent position”. Since then, the central government and local governments in Central China started to pay more attention to the “restrictive policy” of environmental regulation. The elimination of industries with high pollution and high energy consumption is emphasized in important planning and government plans. In 2016, president Xi held a symposium in Chongqing to promote the development of the Yangtze River Economic Belt. He proposed the concept “to step up conservation of the Yangtze River and stop its over development”. Four provinces in Central China are within the Yangtze River Economic Belt. Therefore, most cities in Central China will implement the concept of “ecological priority and green development”. “Restrictive policies” have been given unprecedented importance in Central China. For example, chemical enterprises located within 5 km along the Yangtze River will be forced to “move, transfer or close”. Theoretically, the policy environment in Central China after 2012 has not been very pleasant for enterprises. However, as mentioned above, the local government can also create a pleasant policy environment by giving implicit subsidies and relaxing environmental regulation after 2012. In the following, we quantitatively analyze the spatiotemporal variation of pleasantness of policy environment in Central China.

### 4.2. Estimation of Policy Environment Pleasantness

For enterprises, a pleasant policy environment is composed of encouraging policies and loose restrictive policies. We used the method of Li et al. (2021) and Zhou, Ang, and Poh (2008) to integrate these two types of policies into one indicator based on the entropy method to measure policy environment in prefecture-level cities of Central China [9,38]. The calculation steps of the entropy method are given as follows:(1)$$x_{ij}^{\prime }=\frac{x_{ij}-\min\left(x_{1j},x_{2j},\ldots ,x_{nj}\right)}{\max\left(x_{1j},x_{2j},\ldots ,x_{nj}\right)-\min\left(x_{1j},x_{2j},\ldots ,x_{nj}\right)}+1,\;i=1,\;2,\;\ldots ,\;n;\;j=1,\;2,\;\ldots ,\;m$$
(2)$$x_{ij}^{\prime }=\frac{\max\left(x_{1j},x_{2j},\ldots ,x_{nj}\right)-x_{ij}}{\max\left(x_{1j},x_{2j},\ldots ,x_{nj}\right)-\min\left(x_{1j},x_{2j},\ldots ,x_{nj}\right)}+1,\;i=1,\;2,\;\ldots ,\;n;\;j=1,\;2,\;\ldots ,\;m$$

The first step is to perform nonnegative processing to the data. Equations (1) and (2) adapt for the nonnegative processing of positive indicators and negative indicators separately; *n* represents the number of observations, and *m* represents the number of indicators.
(3)$$p_{ij}=\frac{x_{ij}^{\prime }}{\sum_{i=1}^{n}x_{ij}^{\prime }}$$
(4)$$E_{j}=-\frac{1}{\ln\left(n\right)}\sum_{i=1}^{n}p_{ij}$$

The second step is to calculate information entropy. Equation (4) is the information entropy. Then we can calculate weight value of each indicator:(5)$$W_{j}=\frac{1-E_{j}}{m-\sum_{j=1}^{m}E_{j}}$$

At last, we can calculate the composite index of policy environment:(6)$$S_{i}=\sum_{j=1}^{m}W_{j}*p_{ij}$$

*S_i_* in Equation (6) is the composite index of policy environment. Table 1 shows the construction method of policy environment index, so *m* is 8 and *n* is 1040 in our paper. A higher direct investment and public-service level for a government can make the city more attractive to enterprises and talents, which are positive indicators; on the other hand, loose environmental regulation implies lower pollution-control cost, making cities more attractive to enterprises, so that environmental regulation is a negative indicator. Limited by data availability, the government’s direct investment is measured by general budget expenditure of government and fixed asset investment per unit of GDP. The level of public service is measured by the number of schools, hospitals, and libraries. Environmental regulation is measured by the reciprocal of pollution emission per unit of GDP; pollution emission refers to sulfur dioxide emission, industrial smoke emission, and industrial wastewater emission. Policy environment in this paper is expressed as *POLEN*; in our paper, *POLEN* is *S_i_* in Equation (6). A larger *POLEN* value implies more of a pleasant policy environment for industrial development.

### 4.3. Change and Status of Policy Environment Pleasantness in Central China

Figure 2 shows the change of the average pleasantness of the policy environmental (*POLEN*) in Central China from 2006 to 2018. Besides a brief decline during the financial crisis in 2008, the pleasantness of the policy environment in Central China maintained a rising trend from 2006 to 2016. According to Table 1, a rising trend of pleasantness of the policy environment in Central China might be fostered by higher governments’ investments, better public services (relevant data can be found in China City Statistical Yearbook), and loose environmental regulation. The reason for the decline in 2008 is that governments’ revenue fell due to the impact of the international financial crisis, which led to a decline in expenditure. Even though environmental protection was emphasized after 2012, governments in Central China still created an even more pleasant policy environment for industrial development. There are two reasons for this: (1) After 2012, cities in Central China increased their investments and public-service expenditure. (2) Local governments did not implement the order from the central government to improve environmental regulation. After 2016, the policy environment became less pleasant. As indicated after the symposium held by president Xi, the central government successfully shifted local governments’ focus of Central China from economic growth to environmental protection. Figure 3 shows the spatial variation of policy environment in Central China from 2006 to 2018. Firstly, the pleasantness of the policy environment is obviously spatially heterogeneous. At the provincial level, the policy environment in Henan Province is significantly more pleasant than that in other provinces. The average intensity of environmental regulation in Henan is 0.47, lower than the average value of Central China, which is 0.51. The proportion of government’s expenditure in GDP is 0.176, while the average value is 0.20 in Central China. Thus, the pleasant policy environment in Henan was fostered by its relatively looser environmental regulation. At the city level, the policy environment of provincial capitals is more pleasant than that of other cities. Since provincial capitals often have strict environmental regulations, the main advantage of their policy environment lies in higher public service quality and direct investment in infrastructures. Second, except for Shanxi Province, the pleasant policy environment seems to be continuous. Most cities with a more pleasant policy environment in 2006 will still have a more pleasant policy environment in 2018. In 2006, Shanxi Province took the advantage of the coal industry and achieved rapid economic growth. With sufficient financial funds, Shanxi can provide a pleasant policy environment for the industries. However, the depletion of coal resources and destruction of ecological environment slowed down the development of Shanxi Province; its two regions have been selected as resource-exhausted cities. Therefore, except for Taiyuan, it is difficult for other cities to maintain their attraction to enterprises by creating a pleasant policy environment. From Figure 3, we can also find a phenomenon of spatial agglomeration in cities with higher policy environment pleasantness. That means that there is strong competition among cities in Central China in terms of the policy environment. If a city has a pleasant policy environment, this will cause a positive spillover effect on the policy environment of surrounding cities.

To further clarify the differences of in the policy environment among cities in Central China, we counted the number of cities whose policy environments are more pleasant than the average level of that year (Figure 4). The number of such cities which are defined as cities with more pleasant policy environment is decreasing, indicating that the gap of policy environment pleasantness between cities is becoming larger and the competition intensity in the policy environment between cities is increasing. According to related research studies, industries tend to agglomerate to big cities in Central China, resulting in wider development gaps between cities. When more industries agglomerate to large cities, small cities can no longer afford to provide high-quality public services and investments, thus reducing the number of cities with a more pleasant policy environment. In order to preliminarily explore whether these cities maintain their advantages over other cities through relaxing environmental regulation, we calculated the proportion of cities whose environmental regulations (Calculation method is the reciprocal of pollution emission per unit of GDP; the pollution emission indicators are consistent with Table 1) were lower than the average level of the year in this part of cities. It can be seen that, in most years, cities with relatively lower environmental regulation accounted for more than 50% of cities with a pleasant policy environment. Therefore, we can preliminarily conclude that, in order to maintain the attraction to industries, cities in Central China may create a pleasant policy environment by reducing environmental regulation, resulting in the problem of “regulation capture”, which will have an adverse impact on green growth. Standard econometric methods are adopted to confirm whether improving the pleasantness of the policy environment in Central China is really not conducive to green growth in the following.

## 5. Empirical Tests

### 5.1. Econometric Models

From the above analysis, improving the pleasantness of the policy environment in Central China seems to not be conducive to regional green growth, so we propose Hypothesis 1. 

**Hypothesis** **1** **(H1).**
*Improving the policy environment in Central China will cause a direct negative effect on regional green growth.*


Based on Hypothesis 1, the following econometric model was constructed:(7)$${ENE}_{i,t}=\beta_{0}+\beta_{1}{\mathrm{POLEN}}_{i,t}+{\beta X}_{i,t}+\varepsilon_{i,t}$$
where *ENE* represents green total factor productivity; *POLEN* represents policy environment of cities; ***X*** represents the vector composed of control variables, such as real income level of residents, foreign capital utilization, proportion of manufacturing, natural resources, level of urbanization, and financial support; *i* and *t* are the labels of city units and years, respectively; and *ε_i_*_,*t*_ is the random disturbance.

In addition to the direct impact, a change of policy environment may have indirect impacts on cities’ green growth. Firstly, in order to cope with the competition of cities and promote the coordinated development of cities at different hierarchies within the region, governments in Central China have issued industrial policies and coordination policies, which changed the spatial location of industries and affected the relative industrial scale of cities at different hierarchies. In Central China, provincial capitals have a huge “siphon effect” on the industries of other cities, thus indicating there are only two directions of industrial flow: ordinary cities to provincial capitals and provincial capitals to ordinary cities [3]. Secondly, specialized agglomeration in Central China increased the cost of transformation and limited the ability of creation, so governments in the region have issued a series encouraging policies for the diversified development of industries (These policies can be found on the websites of local governments) [39]. As a result, change of the policy environment suggests change of industrial specialization. Thirdly, to achieve industrial transformation and cope with the competition from eastern developed regions, governments in Central China have issued a series of opening-up stimulus policies and expanded the total foreign trade volume to 343,127 million dollars in 2019, 6.36 times that of 2006 [40]. Lastly, stimulating policies increased the integration level of “industry, education and research” in the region and changed the innovation level of the region [1]. From the existing research, we can find that these changes in Central China may have important impacts on regional green growth. Therefore, in addition to analyzing the direct effect of improving the pleasantness of the policy environment in Central China on green growth, this paper also tries to clarify whether it has indirect effects on green growth.

Here we put forward Hypothesis 2:

**Hypothesis** **2** **(H2).**
*Improving the pleasantness of the policy environment in Central China will have indirect impacts on green growth through affect relative economic scale between cities, industrial structure of cities, technological innovation, and foreign trade.*


To test Hypothesis 2, we further constructed a set of empirical test models:(8)$${\mathrm{MED}}_{i,t}=\beta_{0}+\beta_{1}{\mathrm{POLEN}}_{i,t}+{\beta X}_{i,t}+\varepsilon_{i,t}$$
(9)$${\mathrm{ENE}}_{i,t}=\beta_{0}+\beta_{1}{\mathrm{POLEN}}_{i,t}+\beta_{2}{\mathrm{MED}}_{i,t}+{\beta X}_{i,t}+\varepsilon_{i,t}$$
where the meanings of *ENE*, *POLEN*, and ***X*** are consistent with Equation (7). *MED* represents mediation variables. In order to test Hypothesis 2, *MED* was replaced by the development gap with provincial capital (The industrial flow between ordinary prefecture level cities in Central China is small), industrial specialization, foreign trade, and technological innovation. If *β*_1_ and *β*_2_ in Equation (9) are significant, then *β*_1_ in Equation (8) is significant at the same time, and we can then prove that change of the policy environment in sub-developed regions will have indirect effects on the green total factor productivity. Moreover, *i* and *t* are the labels of city units and years, respectively; and *ε_i_*_,*t*_ is the random disturbance.

### 5.2. Data Analysis

#### 5.2.1. Green Growth in Central China

This paper uses the green total factor productivity to measure the level of green growth. We draw on the Super-SBM (Slack-Based Model) Model of Tone (2001), using non-oriented and non-radical distance function, including undesirable outputs to measure the green economic efficiency of prefecture-level cities in Central China [41]. The model is expressed as follows:(10)$$\min\;\rho =\frac{1-\frac{1}{m}\sum_{i=1}^{m}S_{i}^{-}/x_{ik}}{1+\frac{1}{q}\sum_{r=1}^{q}S_{i}^{+}/y_{ik}}$$
$$s.t.\;x_{k}=X\lambda +s^{-}$$
$$y_{k}=Y\lambda +s^{+}$$
$$\lambda \geq 0,\;s^{+}\geq 0,\;s^{-}\geq 0$$
where *ρ* is green economic efficiency, which is used to indicate the level of green growth; *x_k_* and *y_k_* represent the input and output vectors of DMU, respectively; *X* and *Y* are input-output matrices; *λ* is a column vector with all elements equal to 1; m is the number of input variables; *q* is the number of output variables; and *s*^−^ and *s*^+^ are slack variables. The input variables are capital, labor, and energy utilization. Since prefecture-level cities do not directly report capital stock, capital investment is measured by the value of fixed assets estimated by the perpetual inventory method. Labor input is measured by total employment of prefecture-level cities at the end of the year. The data for energy consumption in prefecture-level cities are not complete, so we used a method to estimate energy input. First, provincial-level data were used to calculate the energy consumption per unit GDP (ten thousand tons of standard coal) of each province, and then the total energy consumption of prefecture-level cities was estimated by multiplying the GDP of prefecture-level cities by energy consumption per unit GDP of each province. The output variables are divided into good output variables (desirable output variables) and bad output variables (undesirable output variables). The good output variable is measured by the real regional GDP of prefecture-level cities. The bad output variable is pollution emission of Prefecture-level cities, which is measured by industrial wastewater emissions, sulfur dioxide emissions, and industrial smoke emissions. In this paper, *ENE* is used to represent green economic efficiency. On the other hand, the total industrial electricity consumption of prefecture-level cities is used to measure energy consumption, and then the green growth index is re-estimated as the dependent variable in the robust test, and it is represented by *ENEA*.

Figure 5 shows the variation of green total factor productivity (*ENE*) in the six central provinces from 2006 to 2018. First, the green total factor productivity in Central China decreased first and then increased. It shows that Central China used to pursue economic growth at the cost of destroying the environment, but environmental-protection pressure and sustainable-growth pressure forced Central China to change its development mode, as well as promote industrial transformation, and then improve the level of green growth. Second, the level of green growth in each province was very low in 2012; however, the green growth level of different provinces were quite different. The green total factor productivity of Anhui was significantly higher than that of other provinces in 2006 and 2018, while the green total factor productivity of Shanxi was always the lowest. This indicates that provinces with a high initial green growth level can quickly transform and return to the green growth path based on early industrial foundation and development experience. It also reflects the development dilemma after the shrinkage of coal industry in Shanxi Province. Third, the green total factor productivity of provincial capitals has been becoming higher than that of other cities in the province.

#### 5.2.2. Mediating Variables: Development Gap with Provincial Capital, Level of Industrial Specialization, Foreign Trade, and Technology Innovation

The change of the relative economic scale of ordinary prefecture-level cities and provincial capital is very conspicuous in Central China. Therefore, the development gap between prefecture-level cities and provincial capital is used to express change of industrial relocation, which is measured by the ratio of GDP of prefecture level city and GDP of provincial capital (*AGG*). A smaller *AGG* represents a larger development gap between prefecture-level cities and provincial capital. Due to high industrial isomorphism in Central China, the change of industrial structure can be described by regional specialization. The relative-diversification index can accurately reflect the industrial specialization of a region. At the same time, the relative-diversification index is more suitable for the horizontal comparison between cities than the absolute-diversification index is [42]; therefore, we estimated the industrial relative-diversification index of the prefecture-level cities in Central China, and it is expressed as *RDI*. We used the ratio of total foreign trade and GDP to measure the status of foreign trade of prefecture-level cities in Central China, excluding the impact of city size on the absolute amount of foreign trade. *FX* is used to represent the status of foreign trade in the cities in Central China. The level of technological innovation is measured by the number of invention patents granted in prefecture-level cities, and it is expressed as *CRE*.

#### 5.2.3. Control Variables

The per capita real income level (*RIC*) of residents is measured by the per capita real wage of prefecture-level cities. The real utilization level of foreign capital (*FDI*) is measured by real foreign direct investment in prefecture-level cities. The proportion of manufacturing (*MAN*) is measured by the proportion of manufacturing employment in total employment of the city. The level of urbanization (*CD*) is measured by the proportion of non-agricultural population in total population of the city. The natural resource (*NR*) value is measured by the proportion of employment in agriculture, forestry, animal husbandry, sideline fisheries, and extractive industries in the total employment of the city; a higher *NR* means that the city has more natural resources [43]. Restricted by the availability of data, we used the year-end loan balance of prefecture-level cities to measure financial support (*FANJR*). Table 2 shows the descriptive statistics of all data.

### 5.3. Benchmark Results

Table 3 reports the test results of whether improving the pleasantness of the policy environment has a negative effect on green growth in Central China. To avoid the influence of heteroscedasticity, we performed logarithmic processing on *RIC*, *FDI*, and *FANJR*. The coefficient of *POLEN* is significantly negative and shows that, from 2006 to 2018, the pleasant policy environment in Central China was not conducive to green growth. Five methods were used to check the robustness of regression results reported in Model (1). Firstly, we excluded the data of six provincial capitals and re-estimated the coefficients of Equation (7). The economic scale of provincial capitals in Central China is so large that the regression results, including the data of provincial capitals, may not reflect the real impacts of changes in the policy environment on green growth within the region. The results of Model (2) show that, after excluding the data of provincial capitals, the coefficient of *POLEN* is still significantly negative and lower than the coefficient in Model (1), indicating that the pleasant policy environment of ordinary prefecture-level cities is more detrimental to green growth. Secondly, we excluded the data before 2010. The regression results of Model (3) show that the coefficient of *POLEN* is significantly negative, indicating that a more pleasant policy environment after 2011 caused higher ecological risks. Thirdly, we replaced the dependent variable *ENE* with *ENEA*. The corresponding regression results are reported in Model (4), showing that the coefficient of *POLEN* is significantly negative. Fourthly, because fixed-effects regression may have endogenous problems, the total number of industrial enterprises above designated size in the prefecture-level cities was selected as the instrumental variable of policy environment, and the instrumental variable method was adopted to estimate the coefficients of Equation (7). A pleasant policy environment intensifies the attraction of a city to enterprises, under which more large-scale enterprises can be cultivated. Therefore, the number of enterprises above the designated size in prefecture-level cities reflects the pleasantness of the policy environment. Theoretically, the number of enterprises does not directly affect environmental efficiency, so this instrumental variable is reasonable. Model (5) reports the corresponding regression results, showing that the coefficient of *POLEN* is still significantly negative. Lastly, the differential GMM regression method is adopted to deal with the endogeneity problems of other variables. The first-order lag term of *ENE* was added to the control variables; At most second-order lag terms of *POLEN*, *CD*, and *MAN* were selected as their own instrumental variables. Model (6) reports the corresponding results, showing that the coefficient of *POLEN* is significantly negative and there are no problems in regard to weak instrumental variables and overidentification. According to the results of the robustness test, it can be concluded that a pleasant policy environment in Central China is not conducive to green growth. The “promotion theory” of policy on green growth is not tenable in Central China.

During this period, economic scale and growth rate of Central China have increased conspicuously, meaning that the decline of green total factor productivity is caused by the increase of pollution emissions. From the results, Central China excessively pursued economic goals, whereas it ignored environmental protection to some extent. Improving the pleasantness of the policy environment should have been “efficient administration” to attract talents and high-quality enterprises to promote industrial transformation and upgrading, but the governments were “captured” under the pressure of short-term economic growth, and then they increased the risks of pollution. As can be seen from Figure 2, besides provincial capitals, the policy environment of a large number of ordinary prefecture-level cities has become more pleasant, in which governments’ intervention has a higher possibility to encourage the development of polluting industries for higher competition pressure and lower development endowments of these cities. In the future, environmental damage risks in ordinary prefecture-level cities in Central China are more likely to be caused by the behavior of local governments.

### 5.4. Mediating Effect Test

Table 4 reports the test results of Hypothesis 2. The regression results of Model (1) and Model (2) show that improving the pleasantness of the policy environment in Central China has widened the development gap between provincial capital and the ordinary prefecture-level city and has had an adverse impact on regional green growth. The negative coefficient of *POLEN* in Model (1) is acceptable. Prefecture-level cities with more pleasant policy environment may bear greater industrial outflow pressure, even improving the pleasantness of policy environment cannot change the “siphon effect” of provincial capitals. Positive coefficient of *AGG* in Model (2) indicates current industrial agglomeration to provincial capitals is conducive to the improvement of green total factor productivity. From the perspective of productivity, industries that move into provincial capitals may improve productivity through positive externality of agglomeration, while industries stay in other cities can also improve productivity by improving the level of specialization. From the perspective of reducing pollution, the relocation of some industries to provincial capitals has reduced the pressure of pollution control in ordinary prefecture level cities and improved the overall efficiency of pollution control in Central China. The results of Model (3) and Model (4) show that improving the pleasantness of the policy environment can indirectly improve green total factor productivity through industrial diversification, indicating. However, this indirect positive effect can be found to be quite low by comparing coefficients of *POLEN* and *AGG* in Model (4). Positive coefficient of *RDI* in Model (3) indicate improving the pleasantness of the policy environment will encourage the development of diversified industries. Positive coefficient of *RDI* in Model (4) implies diversified agglomeration is more beneficial to green innovation through making the innovation chain more complete. Some scholars believe that developing countries have become “pollution refuge paradise” in the process of international trade: With the expansion of foreign trade, the environmental quality will decrease significantly [44]. As the sub-developed region of China, Central China is facing greater competitive pressure, which makes more serious problems of “regulation capture” in its stimulating policies of foreign trade, then the phenomenon of “pollution refuge” will be more prominent in the region. The results of Model (5) and Model (6) show that improving the pleasantness of the policy environment in Central China will have a negative impact on green total factor productivity indirectly through expanding the scale of foreign trade, confirming there is a “pollution refuge” phenomenon in the process of foreign trade within the region. According to the results of Model (7) and Model (8), policy environment in Central China will not have a significant impact on technological innovation, so it will not affect green total factor productivity through this indirect channel. Coefficient of *lnCRE* in Model (8) is negative, which indicates that enterprises in Central China may pay more attention to technological innovation of improving productivity and has ignored environmental protection objectives to some extent.

### 5.5. Heterogeneity Analysis

Cities in Central China have great differences in geographical location, urban characteristics, and development status. By comparing Figure 1 and Figure 2, it can be seen that the spatial changing trail of the pleasantness of the policy environment in Central China is inconsistent with green total factor productivity, indicating that the pleasantness of the policy environment will have heterogeneous effects on the green total factor productivity of different cities. This part will empirically test the heterogeneous impacts of the policy environment on green total factor productivity in Central China from the perspective of spatial location heterogeneity and heterogeneity of characteristics.

#### 5.5.1. Spatial Heterogeneity

Table 5 reports the test results of whether improving the pleasantness of policy environment will have a spatial heterogeneous impact on green total factor productivity. The 33° north latitude line is used to divide Central China into southern part and northern part. The results of Model (1) and Model (2) show that improving the pleasantness of policy environment will cause more adverse effects on green total factor productivity in southern cities, thus indicating that southern cities are more attractive to polluting industries. From Figure 2, we can see that the policy environment of southern cities is becoming more pleasant than that of northern cities, so in the future, southern cities may face more environmental protection pressure than northern cities. Major rivers in Central China are the Yangtze River, the Yellow River, and the Huaihe River. The results of Model (3) and Model (4) report the difference in the impact of improving the pleasantness of policy environment on the green growth of cities along major rivers and cities not along major rivers in Central China. According to the results, improving the pleasantness of the policy environment will have a significant adverse impact on the green total factor productivity of cities not along major rivers. Cities along major rivers always have strict environmental regulations and have performed well in environmental protection so far, so a pleasant policy environment in these cities is not likely to be fostered by relaxing environmental regulations. In order to enhance the competitiveness of cities and introduce positive economies of scale, all provinces in Central China have constructed provincial capital city groups, within which the circulation of production factors and products meets fewer barriers, and the integration level of industries and public services is much higher. We further studied the difference in the impact of improving the pleasantness of the policy environment on green total factor productivity between provincial capital city groups and other areas. Model (5) and Model (6) report the corresponding results. It can be seen that green total factor productivity in cities belonging to provincial capital city groups will decline faster than other cities. On the one hand, with fewer barriers to restrict the mobility of enterprises, industries are more willing to agglomerate to provincial capitals. Ordinary cities in the group are more likely to handle the pressure of economic growth through supporting polluting enterprises, and then they may cause the problem of “environmental regulation capture”. On the other hand, provincial capitals tend to transfer production capacity with high pollution to surrounding cities to achieve industrial upgrading. Thus, provincial capitals will force policy environment of other cities in the group to satisfy polluting enterprises based on their higher position of the hierarchy. Therefore, improving the pleasantness of the policy environment is not conducive to the green growth of ordinary prefecture-level cities belonging to provincial capital city groups.

#### 5.5.2. Heterogeneity of City Characteristics

Table 6 reports the test results of whether improving the pleasantness of policy environment will have heterogeneous impact on green total factor productivity according to the characteristics of cities. First, we divide cities into large cities and small- and medium-sized cities based on the average real GDP of cities located in Central China. According to the results of Model (1) and Model (2), improving the pleasantness of the policy environment has a more significant adverse impact on the green total factor productivity of small- and medium-sized cities. Stronger growth pressure makes small- and medium-sized cities in Central China prefer to achieve short-term growth objectives through encouraging the development of polluting industries, and the pollution-control capacity of these cities is also weaker than that of large cities. Therefore, greater adverse impact on green growth will be caused by governments’ behavior in small- and medium-sized cities. Second, we use the average value of *FX* variable as the standard to divide cities into two categories: cities with a large foreign-trade scale and cities with a small foreign-trade scale. According to the results of Model (3) and Model (4), improving the pleasantness of the policy environment in cities with a larger foreign-trade scale will have a greater adverse impact on green growth. This also confirms the conclusion of the mediating effect test: foreign trade in Central China has the problem of “pollution refuge”; the quality of foreign trade needs to be further improved. Finally, we used the proportion of manufacturing employment to measure the development level of manufacturing and used its average value in Central China as the standard to divide cities into two categories: cities with a high proportion of manufacturing and cities with a low proportion of manufacturing. The results of Model (5) and Model (6) show that a pleasant policy environment in cities with a high proportion of manufacturing in Central China has a greater adverse impact on green growth. Manufacturing is one of the pillar industries in Central China. Different from cities with a large proportion of light manufacturing, in Eastern China, cities in Central China have a large number of enterprises that belong to heavy industries which need huge investment in the early stage, such as equipment manufacturing, steel, automobile, and aviation. Thus, a high transformation cost makes it difficult for such cities to achieve leapfrog development through industrial transformation, and it induces more serious damage to the environment. Moreover, Central China needs to improve the overall operating profits and ecological efficiency of manufacturing in the future.

## 6. Discussion

Whether growth and environmental problems can be solved simultaneously through the intervention of policies has generated hot debates. The regression results in this paper suggest that, when improving the attraction of city to industries, governments in Central China will cause a net negative effect on regional green growth, meaning that governments in sub-developed regions take economic growth more seriously than environmental protection. According to the empirical test results, should cities in Central China eliminate governments’ intervention in industrial development? In this part, we demonstrate that it is very arbitrary to do so from two aspects.

Market mechanism is not necessarily effective in promoting green growth. Price fundamentalism proposes that a price on emissions will reduce emissions more efficiently without the intervention of governments. Proponents of this viewpoint regard pollution as a market failure which will induce negative externalities. Pigouvian tax or purchase on emission right can reduce emissions effectively [45,46]. The US Congressional Budget office estimates a national emissions pricing policy will reduce only 0.03–0.09% GDP loss per annum. However, opponents who solely rely on market mechanism to promote green growth list the evidence that the European Union’s Emissions Trading Scheme has not succeeded in lowering emissions but increased the possibility of corruption (Mufson, S. “Europe’s Problems Color U.S. Plans to Curb Carbon Gases”, 2007.04.09. The Washington Post). Another piece of evidence of market failure is that Latin American countries pursuing neoliberalism have not achieved rapid economic growth, nor have they protected tropical rainforests in the region well. Instead, these opponents have proposed that governments’ behaviors can also effectively reduce pollution emissions, as well as stimulating growth. From the perspective of energy subsidy reform, subsidy removal in fossil fuel has encouraged the transformation of energy industries in India. From the perspective of green industrial policies, encouraging the development of green industries such as wind farms or solar generators will create new jobs in the process of industrial relocation [47,48].

There exists one important reason why sub-developed region such as Central China will not give up the option of the government’s intervention to promote economic growth, even though this will induce pollution: growth might not be easy to achieve without the support from governments. On the one hand, enterprises in Central China often have no comparative advantages for reasons such as relatively shorter survival time, lack of investments, and selling products with low profit margin, so subsidies’ removal will directly reduce their international competitiveness. On the other hand, fierce competition faced by sub-developed region forces local governments to actively participate in economic activities to break through the bottleneck. Green growth requires no substantial reduction in economic growth when improving the ecological environment. Thus, neither a sole market mechanism nor the current governments’ behavior in Central China can achieve this. It is more realistic to avoid regulation capture by improving the quality of administration than eliminating governments’ intervention to achieve green growth in such regions.

## 7. Conclusions

The estimation results based on the data of cities in Central China from 2006 to 2018 show that the pleasantness of the policy environment for industrial development in Central China is increasing, but it contains the risks of damaging the ecological environment. Through empirically testing the relationship between policy environment and green growth, we came to the following conclusions: (1) The current attempts at improving the pleasantness of the policy environment in Central China is not conducive to the green growth of this region. After excluding the data of provincial capitals, changing the time span of the panel data, replacing dependent variable, and performing instrumental variable regression and differential GMM regression, the result is still robust. (2) Improving the pleasantness of the policy environment will have indirect negative impacts on green growth through widening the development gap between ordinary prefecture-level cities and provincial capitals, as well as encouraging foreign trade. A pleasant policy environment will stimulate industrial diversification in Central China, and this will then have an indirect positive effect on green growth. However, the policy environment does not indirectly affect green growth through changing the level of technological innovation. (3) The policy environment in Central China will have a heterogeneous effect on the green growth of cities. From the perspective of spatial heterogeneity, improving the pleasantness of the policy environment in southern cities, cities along major rivers, and cities belonging to provincial capital city groups will have a greater adverse impact on green growth. From the perspective of the heterogeneity of urban characteristics, the adverse impact of improving the pleasantness of the policy environment on green growth is more significant in small- and medium-sized cities, cities with a larger foreign-trade scale, and cities with a high proportion of manufacturing. This paper has two limitations: First, it only identified the possibility of regulation capture in governments’ behavior of Central China, whereas it did not clarify how regulation capture occurred and in which way regulation capture dominant governments’ behavior. Second, the comprehensive tax burden was not considered in estimating the policy environment for lack of data. Next, we, on the one hand, focus on the analysis of regulation capture and, on the other hand, expand data collection to improve the accuracy of policy environment estimated. In order to promote efficient administration, we propose the following:(1)Strengthen environmental regulation. Implement stricter pollution control standards for enterprises. Establish a random inspection mechanism to ensure that enterprises deal with pollution emissions according to the required standards. Recruit professionals to strengthen law enforcement in environmental protection. Through the promotion of environmental supervision APPs on mobile devices, mobilize the public to participate in environmental supervision.(2)Restrict governments’ behaviors to prevent “regulation capture”. Strengthen supervision over officials to prevent bribery of polluting enterprises to corrupt officials. Strengthen the ability of the people’s Congress to identify economic development plans formulated by government, avoiding short-term growth at the expense of environmental damage. Establish a lifelong accountability and punishment mechanism for officials, restrict the investment attraction behavior of officials, and prevent environmental risks caused by preferential policies issued to achieve the goal of investment attraction.(3)Establish a reasonable transfer system on ecological and environmental payments between large cities and small cities to improve overall pollution-control capability in the region. Establish a trading market for pollution-emission rights; cities with serious pollution need to buy pollution-emission rights from cities with less pollution. Economically developed cities shall make financial transfer payments to economically underdeveloped cities. Ordinary cities need to subsidize cities in important ecological regions, because cities in ecological regions bear greater pressure in regard to pollution prevention and control. Encourage enterprises to introduce advanced production equipment and technology.(4)Improve the quality of foreign trade industries and stimulate innovations in environmental protection and pollution control. Cultivate a high-tech foreign trade industry and reduce the number of polluting foreign trade enterprises in the region. Shut down polluting foreign trade enterprises located in forest areas or along major rivers. Subsidize R&D cost of patents for pollution prevention and control and give tax relief to the income of such patents.

## Figures and Tables

**Figure 1 ijerph-19-07647-f001:**
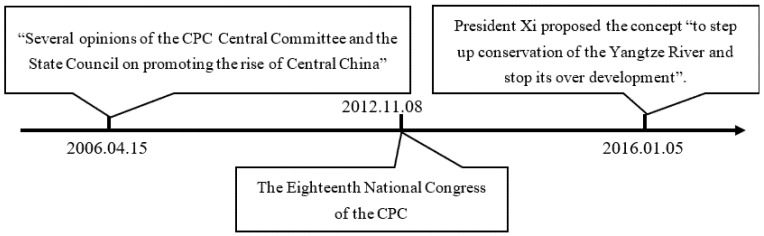
Important events affecting the policy environment in Central China. Note: Built by the authors.

**Figure 2 ijerph-19-07647-f002:**
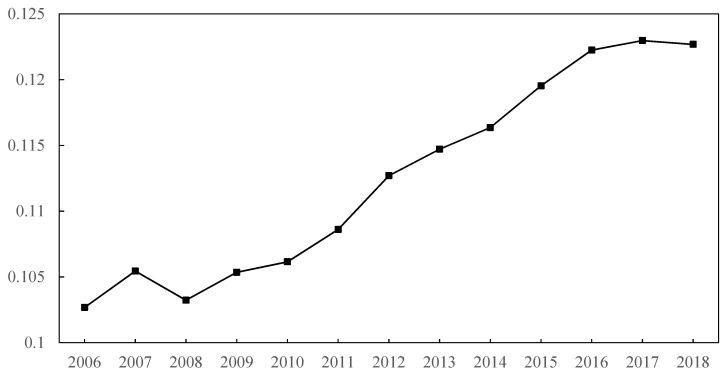
Change of average pleasantness of policy environmental in Central China. Note: Built by the authors based on the data calculated through abovementioned method.

**Figure 3 ijerph-19-07647-f003:**
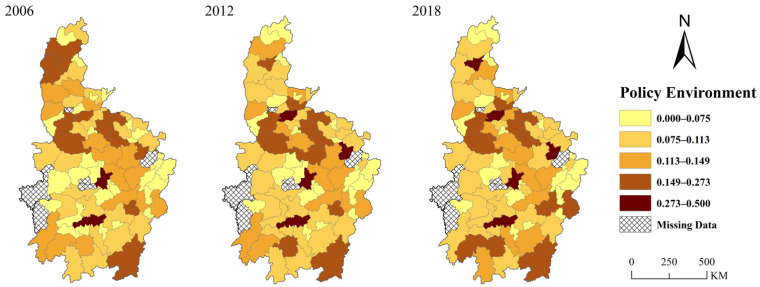
Spatial variation of policy Environment in Central China. Note: Built by the authors based on the data calculated through above method.

**Figure 4 ijerph-19-07647-f004:**
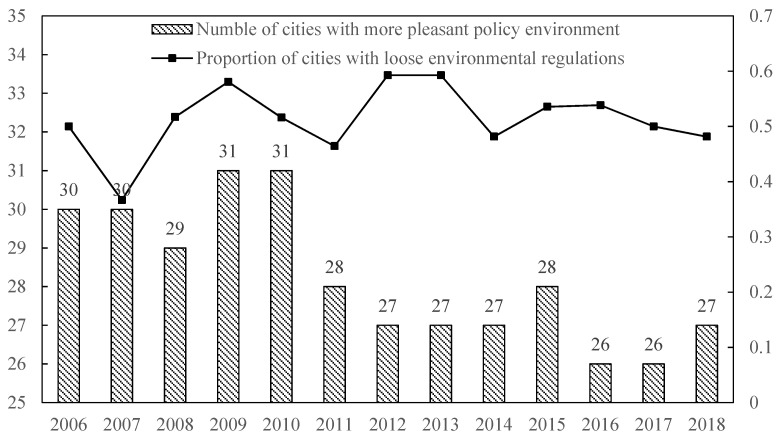
Number of cities with more pleasant policy environment. Note: Built by the authors based on the data calculated through the abovementioned method.

**Figure 5 ijerph-19-07647-f005:**
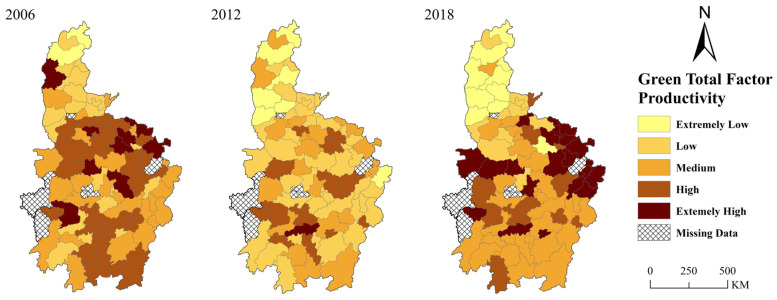
Variation of green total factor productivity in Central China. Note: Built by the authors based on the data calculated through the abovementioned method.

**Table 1 ijerph-19-07647-t001:** Construction of policy environment index.

Primary Indicator	Secondary Indicators	Tertiary Indicators
Policy Environment	Government’s Direct Investment	General Budget Expenditure of Government
		Fixed Asset Investment Per Unit of GDP
	Public Service Level	Number of Schools
		Number of Hospitals
		Number of Libraries
	Environmental Regulation	Sulfur Dioxide Emission
		Industrial Smoke Emission
		Industrial Wastewater Emission

**Table 2 ijerph-19-07647-t002:** Descriptive statistics.

Variable	Obs	Mean	SD	Min	Med	Max
*ENE*	1040	0.522	0.177	0.072	0.495	1
*ENEA*	1040	0.398	0.195	0.013	0.360	1
*POLEN*	1040	0.116	0.081	0.018	0.098	0.564
*AGG*	1040	0.312	0.237	0.017	0.249	1.128
*RDI*	1040	2.640	0.930	0.963	2.499	7.118
*FX*	1040	0.014	0.018	0	0.008	0.216
*CRE*	1040	241.737	716.468	1	51	8797
*ENT*	1040	888.516	543.309	93	777	2968
*RIC*	1040	25,081.170	14,791.960	4009	21,021.830	101,816
*FDI*	1040	56,994.940	95,763.790	1	27,011	1,092,684
*MAN*	1040	0.237	0.106	0	0.229	0.655
*NR*	1040	0.089	0.110	0	0.042	0.543
*CD*	1040	0.280	0.180	0.044	0.228	1
*FANJR*	1040	13,200,000	25,600,000	706,029	6,325,956	268,000,000

**Table 3 ijerph-19-07647-t003:** Benchmark results.

Variables	Model (1)	Model (2)	Model (3)	Model (4)	Model (5)	Model (6)
	*ENE*	*ENE*	*ENE*	*ENEA*	*ENE*	*ENE*
*POLEN*	−1.036 ***	−1.717 ***	−1.914 ***	−1.685 ***	−7.653 *	−1.270 ***
	(−5.58)	(−8.32)	(−8.43)	(−8.55)	(−1.92)	(−36.05)
*LnRIC*	−0.131 ***	−0.122 ***	−0.218 ***	−0.140 ***	−0.082	−0.313 ***
	(−5.08)	(−4.80)	(−6.56)	(−5.10)	(−1.17)	(−58.64)
*LnFDI*	0.008	0.009 *	0.013 *	0.007	0.011	−0.003 *
	(1.57)	(1.73)	(1.87)	(1.31)	(1.31)	(−1.86)
*MAN*	−0.315 ***	−0.358 ***	−0.274 **	−0.021	0.054	−0.514 ***
	(−3.68)	(−4.14)	(−2.18)	(−0.23)	(0.29)	(−18.24)
*NR*	−0.435 ***	−0.625 ***	−0.030	−0.381 **	−0.539 *	−0.262 ***
	(−2.83)	(−4.19)	(−0.12)	(−2.33)	(−1.92)	(−6.83)
*CD*	0.223 ***	0.133 *	0.218	0.087	0.186 *	0.146 ***
	(3.05)	(1.76)	(1.64)	(1.12)	(1.71)	(11.30)
*LnFANJR*	−0.014	−0.022	0.094 ***	−0.010	−0.005	0.070 ***
	(−0.98)	(−1.59)	(4.59)	(−0.66)	(−0.15)	(21.13)
*L.ENE*						−0.061 ***
						(−8.36)
*Constant*	2.144 ***	2.258 ***	1.302 ***	2.089 ***	2.099 **	2.860 ***
	(15.73)	(17.05)	(4.55)	(14.45)	(2.07)	(102.81)
Regression Method	Fixed Effects	Fixed Effects	Fixed Effects	Fixed Effects	Fixed Effects IV	Difference GMM
R-squared	0.236	0.299	0.252	0.264		
Arellano–Bond test AR (1)						0
Arellano–Bond test AR (2)						0.395
Sargan test						0.859
Observations	1040	962	640	1040	1040	880

Note: t- or z-statistics in parentheses; *** *p* < 0.01, ** *p* < 0.05, and * *p* < 0.1.

**Table 4 ijerph-19-07647-t004:** Mediating Effect Test Results.

**Panel A**				
**Variables**	**Model (1)**	**Model (2)**	**Model (3)**	**Model (4)**
	** *AGG* **	** *ENE* **	** *RDI* **	** *ENE* **
*POLEN*	−1.057 ***	−0.596 ***	2.396 ***	−1.093 ***
	(−9.56)	(−3.16)	(2.90)	(−5.89)
*AGG*		0.416 ***		
		(7.89)		
*RDI*				0.024 ***
				(3.29)
Controls	Yes	Yes	Yes	Yes
Regression Method	Fixed Effects	Fixed Effects	Fixed Effects	Fixed Effects
R-squared	0.392	0.493	0.235	0.283
Observations	1040	1040	1040	1040
**Panel B**				
**Variables**	**Model (5)**	**Model (6)**	**Model (7)**	**Model (8)**
	** *FX* **	** *ENE* **	** *LnCRE* **	** *ENE* **
*POLEN*	0.201 ***	−0.636 ***	0.063	−1.035 ***
	(13.02)	(−3.20)	(0.08)	(−5.59)
*FX*		−1.993 ***		
		(−5.18)		
*LnCRE*				−0.021 ***
				(−2.73)
Controls	Yes	Yes	Yes	Yes
Regression Method	Fixed Effects	Fixed Effects	Fixed Effects	Fixed Effects
R-squared	0.268	0.257	0.808	0.242
Observations	1040	1040	1040	1040

Note: t- or z-statistics in parentheses; *** *p* < 0.01.

**Table 5 ijerph-19-07647-t005:** Heterogeneity test results A: spatial heterogeneity.

Variables	Model (1)	Model (2)	Model (3)	Model (4)	Model (5)	Model (6)
	Northern Area		Along Major Rivers		Capital City Group	
	Yes	No	Yes	No	Yes	No
	*ENE*	*ENE*	*ENE*	*ENE*	*ENE*	*ENE*
*POLEN*	−0.533 **	−2.360 ***	−0.565	−1.393 ***	−1.185 ***	−0.963 ***
	(−2.29)	(−7.56)	(−1.63)	(−6.31)	(−5.30)	(−3.32)
*LnRIC*	−0.161 ***	−0.041	−0.140 ***	−0.125 ***	−0.149 ***	−0.192 ***
	(−4.45)	(−1.11)	(−3.15)	(−3.84)	(−3.54)	(−5.52)
*LnFDI*	0.015 **	−0.007	0.008	0.007	0.008	0.024 *
	(2.21)	(−0.79)	(0.58)	(1.29)	(1.44)	(1.74)
*MAN*	−0.299 **	−0.202 *	−0.422 ***	−0.257 **	−0.480 ***	−0.151
	(−2.50)	(−1.66)	(−2.76)	(−2.48)	(−3.53)	(−1.39)
*NR*	0.360	−0.882 ***	0.182	−0.594 ***	−0.874 ***	−0.043
	(1.34)	(−4.83)	(0.50)	(−3.51)	(−4.10)	(−0.20)
*CD*	0.268 ***	0.241 **	0.241 **	0.236 **	0.095	0.281 ***
	(2.67)	(2.31)	(2.12)	(2.44)	(0.60)	(3.35)
*LnFANJR*	0.016	−0.063 ***	0.016	−0.030 *	−0.049 ***	0.035
	(0.79)	(−3.12)	(0.50)	(−1.77)	(−2.68)	(1.44)
*Constant*	1.791 ***	2.272 ***	1.723 ***	2.343 ***	2.997 ***	1.755 ***
	(8.80)	(12.75)	(6.85)	(14.10)	(12.37)	(9.81)
Regression Method	Fixed Effects	Fixed Effects	Fixed Effects	Fixed Effects	Fixed Effects	Fixed Effects
R-squared	0.205	0.336	0.172	0.297	0.391	0.205
Observations	494	546	390	650	364	676

Note: t- or z-statistics in parentheses; *** *p* < 0.01, ** *p* < 0.05, and * *p* < 0.1.

**Table 6 ijerph-19-07647-t006:** Heterogeneity test results B: heterogeneity of city characteristics.

Variables	Model (1)	Model (2)	Model (3)	Model (4)	Model (5)	Model (6)
	City Size		Foreign Trade		Proportion of Manufacturing	
	Large	Small and Medium	High	Low	High	Low
	*ENE*	*ENE*	*ENE*	*ENE*	*ENE*	*ENE*
*POLEN*	0.393	−1.762 ***	−0.876 **	−0.775 **	−1.457 ***	−1.008 ***
	(0.97)	(−8.26)	(−2.14)	(−2.32)	(−4.97)	(−3.87)
*LnRIC*	−0.073	−0.112 ***	−0.177 ***	−0.083 ***	−0.064 *	−0.158 ***
	(−1.00)	(−4.17)	(−3.07)	(−2.74)	(−1.68)	(−4.42)
*LnFDI*	−0.0452 *	0.006	0.035	0.006	0.019	0.007
	(−1.92)	(1.24)	(1.49)	(1.10)	(1.37)	(1.21)
*MAN*	−0.229	−0.234 **	0.121	−0.438 ***	−0.426 ***	−0.095
	(−1.35)	(−2.30)	(0.67)	(−4.09)	(−2.91)	(−0.51)
*NR*	0.574	−0.580 ***	0.110	−0.525 ***	0.585 *	−0.486 **
	(0.99)	(−3.73)	(0.34)	(−2.99)	(1.95)	(−2.33)
*CD*	0.689 ***	0.107	0.414 **	0.106	0.137	0.120
	(3.92)	(1.37)	(2.51)	(1.28)	(1.40)	(0.98)
*LnFANJR*	−0.015	−0.042 ***	0.043	−0.040 **	−0.011	−0.016
	(−0.30)	(−2.78)	(1.10)	(−2.47)	(−0.47)	(−0.84)
*Constant*	1.843 ***	2.433 ***	1.205 ***	2.117 ***	1.411 ***	2.400 ***
	(3.99)	(16.64)	(2.85)	(14.47)	(5.62)	(13.34)
Regression Method	Fixed Effects	Fixed Effects	Fixed Effects	Fixed Effects	Fixed Effects	Fixed Effects
R-squared	0.132	0.376	0.118	0.206	0.215	0.244
Observations	319	721	305	735	493	547

Note: t- or z-statistics in parentheses; *** *p* < 0.01, ** *p* < 0.05, and * *p* < 0.1.

## Data Availability

Data sources are China City Statistical Yearbooks, China Statistical Yearbooks, Statistical Yearbooks of six provinces in Central China, and Statistical Bulletins of each city from 2007 to 2019. Data of patents are from CNRDS database and the official website of National Bureau of Statistics.

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
