# Peer review of "Is a Pleasant Policy Environment Conducive to Green Growth of Central China?"

_ijerph, 2022, doi:10.3390/ijerph19137647_

Round 1

Reviewer 1 Report

This work is potentially interesting but very confusing and methodologically weak. The authors introduce variables without explanations. The authors use the concept of green growth as a given without considering the debate over green growth. 

Author Response

Response to reviewer 1's comments:

For the comments of reviewer 1, we carefully examined the structure, methods and contents of the paper. From the perspective of structure, this paper is a standard empirical research paper without much confusion. The structure is divided into three parts: research background and objectives, research hypotheses and research methods, data and empirical tests. From the perspective of research methods, we used the methods of data visualization analysis, OLS regression, IV regression, GMM regression, mediation effect test and heterogeneity analysis, which are common empirical research methods. From the perspective of contents, we think it may be description of research objectives and estimation method of some variables are not clear enough. So, we made the following adjustments.

(1) In the introduction part, research objectives and research methods are further descripted.

(2) Although the estimation methods of all important variables and where these methods come from have been described in detail, the reviewers still proposed the problem of lack of "explanation on variables". After careful examination, we decided to further explain the estimation methods of RDI and NR and we added references of the estimation methods of these two variables.

(3) As for reviewer proposed that “use the concept of green growth without considering the debate over green growth”, based on the fact that the focus of this study is whether improving pleasantness of policy environment in Central China is conducive to green growth, we believe that we haven’t missed important information about green growth. At present, there is little debate over the definition of green growth, whereas most debates lie in the field of whether environmental protection and growth can be achieved at the same time, and what factors affect green growth. We have listed these debates in “2.2. Researches on green growth of China” and “2.3. Researches on the relationship between policy environment and green growth”. And we have also analyzed the relevant researches on the definition and measurement of green growth, then compared the estimated results of different researchers. In the part of “5.2.1. Green growth in Central China”, we have not only explained the estimation method of green growth in this paper, but also analyzed its spatiotemporal variation in Central China. Nevertheless, we have added a discussion section, in which we listed the current debates on whether government’s behavior or market mechanism is more suitable for promoting green growth and discussed the possible options for Central China.

(4) English writing has been checked and improved.

Reviewer 2 Report

Green growth is a topical issue and is well discussed and discussed in the article.

The title of the article is well-chosen given the current background of green growth.

The abstract gives a succinct presentation of the content of the article and is well-argued in terms of the topic.

Generally speaking, the article has a good structure and presents the research methodology and objectives in a clear manner.

The obtained results are generally presented in detail, with pertinent value judgments, in accordance with the data collected and the processing considered.

The conclusions are well structured and present the implications of the analysis and the proposed policies.

I recommend adding some new references, from the last few years, to support your hypotheses.

Author Response

Dear reviewer 2:

We rewrote the part of research hypotheses and listed the relevant literatures on the impact of government’s behavior on regional industrial relocation, industrial structure adjustment, foreign trade and technological innovation to support our research hypotheses.

Reviewer 3 Report

I thank the editors for the opportunity to read and comment on this paper. Theoretical and empirical analysis of the advantages of open and competitive economies is critical today, where omnipotent governments try to direct the lives of individuals. While the former involves efficient management, the latter involves regulation capture. However, the analyzed paper does not offer new evidence on how to avoid regulation capture. On the contrary, it offers more additional mechanisms for government intervention, which increases the regulation capture.

Remarkably, the authors overlook the theoretical findings of the Austrian school of economics and the public choice school on the problems of central planning in economic and environmental affairs. In this sense, the authors should include the essence of sustainable growth and development and the spontaneity of the market process.

Some improvement suggestions:

- Better explain the hypothesis and objectives of the paper in the summary and introduction and how you will achieve your objectives.

- Lines 45-48 seem to be missing the quote on the definition of growth and sustainable development.

- Lines 72-81 seem to be missing the quote.

- Lines 186-202 should be in the introduction because it explains the hypotheses and objectives of the paper.

- Tables and figures are not cited. Where did you get the data from? Were the tables built by yourselves or copied from other sources?

- Table 1 is not adequately explained. Who created the policy environment index? Who uses it? Are these variables enough to avoid regulation capture?

- Figures 1, 2, 3 and 4 are not adequately explained. Some data are not commented.

- If the model shows how government intervention has damaged the environment, why do the authors propose more government restrictions? Alternative options and how the proposals effectively eliminate regulation capture are not mentioned. Are there other alternatives consistent with free enterprise and the market economy?

- The paper is a bit scattered. I suggest orienting the entire paper to the subject of capture regulation.

Here are some bibliographical references:

- Espinosa, V. I., Alonso Neira, M. A., & Huerta de Soto, J. (2021). Principles of sustainable economic growth and development: A call to action in a post-COVID-19 world. Sustainability, 13(23), 13126.

- Huerta de Soto, J. (2010). Socialism, economic calculation, and entrepreneurship. Northhampton, MA: Edward Elgar.

- Block, W. (1998). Environmentalism and Economic Freedom: The Case for Private Property Rights. Journal of Business Ethics, 17(16), 1887-1899.

- Block, W. (1990). Economics and the Environment: a Reconciliation. Fraser Institute.

- Kirzner, I.M. (2017). The entrepreneurial market process—An exposition. Southern Economic Journal, 83(4), 855-868.

- Espinosa, V. I., Peña-Ramos, J. A., & Recuero-López, F. (2021). The political economy of rent-seeking: Evidence from Spain's support policies for renewable energy. Energies, 14(14), 4197.

- Anderson, T., & Leal, D. (Eds.). (2015). Free market environmentalism for the next generation. Springer. 

Best, 

Reviewer. 

Author Response

Dear reviewer 3:

In essence, the purpose of this article is not to advocate liberalism or deny the value of government’s intervention. This article is only an empirical analysis to evaluate the administrative efficiency of government in Central China. Therefore, the reviewer may have some misunderstandings about the research objectives and the meaning of the key variable (POLEN). Firstly, although policy environment is directly affected by government’s intervention, it is not equal to “government direct lives of individuals”. Even in the United States, we can still judge the differences in pleasantness of policy environment for industrial development in different states for they are with different comprehensive tax burden. Change of policy environment may have a certain impact on green growth. For emerging economies such as China, governments’ intervention is more active, resulting in a complex composition of policy environment. So, we think it is very important to study whether the change of policy environment in Central China is conducive to green growth. Secondly, the viewpoint “Open economies involves efficient management, governments’ intervention involves regulation capture” of the reviewer is not neutral and rigorous. Market mechanism always fails especially in the field of environmental protection, examples can be found in protection of tropical rain forests in Latin American countries. Also, government’s support in sub-developed region is important for enterprises to cope with competitions from developed regions. Therefore, we cannot arbitrarily conclude that competitive economy must be better than economy with government’s intervention. On the contrary, based on the data of central China, this paper objectively evaluates the efficiency of improving pleasantness of policy environment from the perspective of regional green growth. Based on this, we put forward suggestions to reduce the distortions of government’s intervention, rather than reduce government’s intervention. In order to reduce the misunderstanding of readers, we have made the following adjustments.

(1) We have re-written the introduction and clearly put forward the definition of policy environment at the beginning of the introduction. In the introduction, objective of the research is further clarified: Empirically tests the relationship between improving pleasantness of policy environment and green growth. Then we clarified our research methods and the structure of the full text.

(2) Endnotes have been added to explain some special concepts. There are 11 endnotes total in the full text.

  (3) Contents of the literature review have been adjusted: Research objective is moved to the introduction.

(4) The contents of “3. Data Sources” have been expanded: The data used in this paper are described in detail.

(5) Notes have been added to figures 2, 3, 4 and 5 to explain that the picture is drawn by the authors and inform readers of the calculation methods of data used in these figures. Data source of the figures is the same with contents of “3. Data Sources”. Figure 1 is a description of the events, which does not involve the use of data, so it is only noted with “Built by the authors”. We have consulted with relevant experts that tables used for empirical analysis do not involve contents of other literatures, so it is not needed to cite it. For details, the reviewer can refer to the papers published in AER or QJE.

(6) We have further explained the estimation method of policy environment in Table 1. We added references of the estimated method. We did not create this indicator to avoid regulatory capture, but only to measure whether governments’ behavior was friendly to the development of enterprises. Entropy method can be used to integrate similar indicators into a comprehensive indicator. This method is widely used in geography and public management. Here, we recommend to the reviewer a paper "Study on classification and applicability of comprehensive evaluation methods" written by Zhang Xia and He Nan which has been published in “Statistics & decision” in 2022.

(7) Figures 2, 3 and 4 have been further explained, especially explained the causes of some important features. Figure 1 is only a description of the events. We believe that the contents describe Figure 1 has fully expressed their background, process and impact on the pleasantness of policy environment.

(8) We added a new discussion section to reduce the misunderstanding of readers. The reviewer feels confused about “since governments’ intervention in Central China is not conducive to green growth, why it can’t be reduced”. In this part, we demonstrate it is very arbitrary to just reduce the interventions of governments. We have added current discussions on the impact of market mechanism and government’s intervention on green growth. It can be found that market mechanism will not necessarily promote green growth and not all government’s interventions are not conducive to green growth. On the other hand, we have explained the importance of government’s intervention for economic growth of sub-developed regions, government’s intervention in such regions can’t be simply eliminated. Therefore, our proposal is by no means as simple as increasing or reducing government’s intervention: we suggest increasing government’s intervention that is conducive to green growth and restricting government’s behavior that is not conducive to green growth.

(9) As for the reviewer's suggestion that we should change the subject of this paper into studying the impact of regulation capture, we believe that it is neither necessary nor easy to operate. Policy environment is not equal to regulation capture. Similarly, the impact of policy environment on green growth is not equal to the impact of regulation capture on green growth. Figure out whether regulation capture is possible is only one of the significances of studying relationship between pleasant policy environment and green growth. Regulation capture is difficult to be observed directly, just as it is impossible for us to directly observing officials to participate in corrupt activities. So, we can only identify the possible regulation capture risks through certain empirical analysis. In the next step of the research, we will obtain more evidence about regulation capture to further explain the causes and mechanisms of regulation capture.

Reviewer 4 Report

I believe that the article presented for review is important for the sustainable development of China, but not only. The conclusions of the research can be used to formulate strategies and policies in other regions of the world.

What deserves attention is the fact that reading this manuscript one can notice a very thoughtful and logical structure of the argument.

All the elements characteristic of a scientific paper have been fully taken into account. The authors have even included a section devoted to comments on existing researches.

The time span of the research covers 13 years. The literature review is based on 40 sources. 

In the section on research methodology, the authors have explained each formula very well, making the proposed empirical model fully readable (transparent). Moreover, it should be emphasized that the authors used advanced econometric methods.

The results section states the findings of the research arranged in a logical sequence.

In the conclusions section, the authors formulate recommendations for policy. However, I would suggest numbering the lines here: 639, 645, 652, 660. It would also be appropriate for the conclusions to indicate the limitations of the analyses and possible directions for further research.

Reassuming, I believe that the presented article meets all the requirements for scientific studies. After taking into account the reported comments, I recommend it for publication.

Author Response

Dear reviewer 4:

In the conclusion part, we add two limitations of this research and next research directions.

Round 2

Reviewer 1 Report

For me it remains a confusing paper. For example, there is no definition of the entropy method except a reference to two papers. But that not sufficient they should say how they implement the method. Green Growth is not a given. There is now a serious debate whether green growth is doable with today technology. Green growth requires decoupling of growth from use of natural resources an objective probably not attainable with current technologies. 

Anyway, the paper is full of data and might be useful for the Chinese environment. I change my evaliuation to accept

Author Response

Dear reviewer 1:

We are thankful to your suggestions on modifying  the paper.  Modifications have been made as follow:

(1) Title have been changed to “Is Pleasant Policy Environment Conducive to Green Growth of Central China?”. The reason for the change is that the previous title is easy to misunderstand readers that the focus of this paper is regulation capture. The title we changed makes it easier for readers to find that the research focus of this paper is to empirically test the impact of policy environment change on green growth in Central China.

(2) We have re-written the “Introduction”. The new introduction highlights the research background, research necessity and urgency, research methods, and clarifies the research objective is to study the impact of policy environment changes on regional green growth. The logic of the introduction is: The first paragraph defines policy environment, explains the importance of changes in the pleasantness of policy environment to the economic growth of Central China, and states the background of current return of policy support in the region to achieve the goal of “the rise of Central China”; The second paragraph explains the environmental risks involved in current improving pleasantness of policy environment in Central China from the perspective of encouraging policies and restrictive policies, and brings out the problem that pleasant policy environment may not be conducive to regional green growth; The third paragraph illustrates the urgency and necessity of studying the impact of policy environment on green growth in Central China by stating the realistic environmental risks faced by Central China and the lack of relevant researches; The fourth paragraph puts forward that the research objective of this paper is "to empirically clarify whether and how improving pleasantness of policy environment in Central China affects regional green growth, then provide useful policy implications for sub-developed regions to reduce adverse effects of administration on environmental protection and explore a green growth path" and introduces the research methods of this paper. In the "Comments on existing researches", we proposed the "innovation" of this paper, because we think it is not rigorous to put forward the innovation without reviewing the relevant literature.

(3) In order to ensure that the research content does not deviate from the theme of the impact of policy environment on green growth, we have adjusted the content of the paper. For example, expressions as "regulation capture improved the pleasantness of policy environment then caused adverse impact on green growth" in the “Abstract”, “Empirical Tests” and “Conclusions” have been removed, and the impact of government’s intervention on green growth is explained from the perspective of policy environment change.

(4) Figures 1, 2, 3 and 4 are explained more accurately and completely. Figures 1, 2, 3 and 4 in this paper are built to reflect changes of policy environment pleasantness rather than changes of "regulation capture", so we reinterpret these figures from the perspective of encouraging and restricting policies introduced by governments. As for figure 1, we use " However, as mentioned above, local government can also create pleasant policy environment by giving implicit subsidies and relaxing environmental regulation after 2012. " to replace the original expression. For figure 2, we have further analyzed the change trend and the reasons behind it. For figure 3, we have explained causes for characteristics of the policy environment in Henan Province. For figure 4, we have explained the downward trend and used "relaxing environmental regulation" to substitute "regulation capture".

(5) We have added endnotes to some important concepts to reduce readers' confusions.

(6) We have detailed measurement method of important data in the paper: we have detailed the entropy method which is used to estimate the pleasantness of policy environment and the SBM model which is used to estimate the level of green growth.

Reviewer 3 Report

The paper has new sentences to clarify some points, but does not address the substance of the Round 1 comments.

For example:

- The novelty of the thesis and findings of the paper are not adequately explained.

- Alternative approaches to their thesis that "economic growth cannot occur without government intervention" are not explained.

- They do not explain how government intervention, and the paper proposals, create "regulation capture", corruption effect, dynamic inefficiency, etc.

- The quantitative results of the figures are not adequately explained.

Author Response

Dear reviewer 3:

Thank you for pointing out some important modifications needed in the paper. We have thoughtfully taken into account these comments. 

Through a careful study on the suggestions, we have found that it is inaccurate to attribute the adverse impact of policy environment changes on green growth of Central China to regulation capture in our early writing of the paper, as a result we ignored the analysis of policy environment changes to some certain extent. In fact, the risk of policy environment change to regional environmental protection is not only from "regulation capture", but also from encouraging policies. In the former analysis, the problem of "regulation capture" was too much highlighted, which misled our objective from clarifying whether improving pleasantness of policy environment in Central China is conducive to regional green growth to study the relationship between “regulation capture” and green growth. The pleasantness of policy environment is fostered by government's encouraging and restrictive policies. Analyzing whether and how policy environment change affects green growth can more accurately grasp the relationship between government’s intervention and green growth in the region. In order to highlight the research objectives of this paper, focus the analysis on explaining whether improving pleasantness of policy environment is conducive to the green growth of Central China, and provide more sufficient evidence to support our analysis, modifications have been made as follow:

(1) Title have been changed to “Is Pleasant Policy Environment Conducive to Green Growth of Central China?”. The reason for the change is that the previous title is easy to misunderstand readers that the focus of this paper is regulation capture. The title we changed makes it easier for readers to find that the research focus of this paper is to empirically test the impact of policy environment change on green growth in Central China.

(2) We have re-written the “Introduction”. The new introduction highlights the research background, research necessity and urgency, research methods, and clarifies the research objective is to study the impact of policy environment changes on regional green growth. The logic of the introduction is: The first paragraph defines policy environment, explains the importance of changes in the pleasantness of policy environment to the economic growth of Central China, and states the background of current return of policy support in the region to achieve the goal of “the rise of Central China”; The second paragraph explains the environmental risks involved in current improving pleasantness of policy environment in Central China from the perspective of encouraging policies and restrictive policies, and brings out the problem that pleasant policy environment may not be conducive to regional green growth; The third paragraph illustrates the urgency and necessity of studying the impact of policy environment on green growth in Central China by stating the realistic environmental risks faced by Central China and the lack of relevant researches; The fourth paragraph puts forward that the research objective of this paper is "to empirically clarify whether and how improving pleasantness of policy environment in Central China affects regional green growth, then provide useful policy implications for sub-developed regions to reduce adverse effects of administration on environmental protection and explore a green growth path" and introduces the research methods of this paper. 

(3) In order to ensure that the research content does not deviate from the theme of the impact of policy environment on green growth, we have adjusted the content of the paper. For example, expressions as "regulation capture improved the pleasantness of policy environment then caused adverse impact on green growth" in the “Abstract”, “Empirical Tests” and “Conclusions” have been removed, and the impact of government’s intervention on green growth is explained from the perspective of policy environment change.

(4) Figures 1, 2, 3 and 4 are explained more accurately and completely. Figures 1, 2, 3 and 4 in this paper are built to reflect changes of policy environment pleasantness rather than changes of "regulation capture", so we reinterpret these figures from the perspective of encouraging and restricting policies introduced by governments. As for figure 1, we use " However, as mentioned above, local government can also create pleasant policy environment by giving implicit subsidies and relaxing environmental regulation after 2012. " to replace the original expression. For figure 2, we have further analyzed the change trend and the reasons behind it. For figure 3, we have explained causes for characteristics of the policy environment in Henan Province. For figure 4, we have explained the downward trend and used "relaxing environmental regulation" to substitute "regulation capture".

(5) We have added endnotes to some important concepts to reduce readers' confusions.

(6) We have detailed measurement method of important data in the paper: we have detailed the entropy method which is used to estimate the pleasantness of policy environment and the SBM model which is used to estimate the level of green growth.

Explanations to your questions:

(1) In the "Comments on existing researches", we proposed the "innovation" of this paper, because we think it is not rigorous to put forward the innovation without reviewing the relevant literatures.

(2) Your expression “economic growth cannot occur without government intervention” is not the same with relevant expressions in our paper, such as “Policy environment, which refers to whether government’s intervention in a region is friendly to the operation of enterprises, in Central China is an important factor affecting industrial development”, “Since governments’ intervention is an un-substitutable driving force for the growth of Central China in the short-term”, “Growth might not be easy to achieve without the support from governments”. We explained this through adding endnote 7 and supplementing content in “Discussion”: “On the one hand, enterprises in Central China often have no comparative advantages for reasons such as relatively shorter survival time, lack of investments and selling products with low profit margin, subsidies removal will directly reduce their international competitiveness. On the other hand, fierce competition faced by sub-developed region forces local governments actively participate in economic activities to break through the bottleneck”; “Industries in China tend to agglomerate to economically developed regions, if there is no intervention from governments in Central China, development gap will be wider between Central China and Eastern China. Industries in Central China are those with heavy in-vestments and long return cycle, which is less competitive compared to IT or financial industries mostly located in eastern large cities, making them more reliance on policy sup-port [41]”.

(3)  We believe that regulation capture can explain only a part of adverse impact of pleasant policy environment on green growth. Therefore, our research objective is clearly defined as the impact of policy environment on green growth, the regulation capture is not our research focus. For the reviewer’s question on how government’s intervention causes "regulatory capture", we have given a simple explanation in the new introduction: “That means governments in sub-developed regions such as Central China may actively or passively "collaborate" with polluting industries due to fierce regional competition and huge pressure of growth, causing the problem of ‘environmental regulation capture’. On the other hand, slow reform of the administrative system in Central China results in defects of supervision over bureaucratic system, making it easier to induce environmental corruptions”.

(4) We have described main features of figures and explained the obvious reasons according to the factors affecting the pleasantness of policy environment in Table 1. We believe that the figures should not be over explained: There are many factors affecting the change of policy environment, detail subtle characteristics is not the main purpose of this paper; At the same time, the explanation without support of empirical research may not be accurate.